# The role of MHC class I recycling and Arf6 in cross-presentation by murine dendritic cells

Sebastian Montealegre[1,2,3] , Anastasia Abramova[1,2,3], Valerie Manceau[1,2,3], Anne-Floor de Kanter[1,2,3] , Peter van Endert[1,2,3]

**Cross-presentation by MHC class I molecules (MHC-I) is critical for priming of cytotoxic T cells. Peptides derived from cross-presented antigens can be loaded on MHC-I in the endoplasmic reticulum and in endocytic or phagocytic compartments of murine DCs. However, the origin of MHC-I in the latter compartments is poorly understood. Recently, Rab22-dependent MHC-I recycling through a Rab11+ compartment has been suggested to be implicated in cross-presentation. We have examined the existence of MHC-I recycling and the role of Arf6, described to regulate recycling in nonprofessional antigen presenting cells, in murine DCs. We confirm folded MHC-I accumulation in a juxtanuclear Rab11+ compartment and partially localize Arf6 to this compartment. MHC-I undergo fast recycling, however, both folded and unfolded internalized MHC-I fail to recycle to the Rab11+Arf6+ compartment. Therefore, the source of MHC-I molecules in DC endocytic compartments remains to be identified. Functionally, depletion of Arf6 compromises cross-presentation of immune complexes but not of soluble, phagocytosed or mannose receptor–targeted antigen, suggesting a role of Fc receptor–regulated Arf6 trafficking in cross-presentation of immune complexes.**

## Introduction

MHC class I molecules (MHC-I) mainly present peptides derived through the degradation of intracellular proteins to CTL, using the so-called direct antigen presentation pathway. In specialized or professional APCs including foremost DCs, peptides derived from extracellular antigens can also be loaded onto MHC-I in a process known as cross-presentation (Alloatti et al, 2016). Both types of antigen presentation are fundamental processes in the defense against pathogens and tumors.

Work on nonprofessional APCs has shown that upon arrival to the cell surface, MHC-I can divide into different membrane domains according to their peptide-loading status (Mahmutefendić et al, 2011), from where they are constantly internalized to endosomal compartments in a clathrin-independent manner (Eyster et al, 2009; Montealegre & van Endert, 2018). In such cell lines, MHC-I can recycle to the cell surface, in a process regulated by the small GTPases Arf6 (Radhakrishna & Donaldson, 1997; Jovanovic et al, 2006), Rab22 (Weigert et al, 2004) and the epsilon homology domain proteins 1 and 3 (EHD-1 and EHD-3). Whether class I molecules are recycled or targeted to lysosomal degradation depends on the affinity of the peptide bound and on the association with $\beta_2$-microglobulin ($\beta_2$m). Whereas peptide-bound class I molecules can recycle from an early endosome (Zagorac et al, 2012), once $\beta_2$m has dissociated from the MHC-I heavy chain (HC), the vast majority become targeted to degradation in the lysosomes (Montealegre et al, 2015), although a late endosomal recycling pathway has been reported (Mahmutefendić et al, 2017).

Cross-presentation is thought to use multiple pathways that can implicate peptide loading of MHC-I in several intracellular environments, including the perinuclear ER, specialized compartments formed by fusion of the ER with phagosomes or endosomes, and "vacuolar" late endosomes/lysosomes (Guermonprez et al, 2003; Shen et al, 2004; Burgdorf et al, 2008; Cruz et al, 2017). However, the source of MHC-I in the latter two pathways remains obscure. In principle, MHC-I could be recruited to endocytic compartments through recycling, from the secretory pathway or potentially as newly synthesized molecules bypassing the secretory pathway (Ma et al, 2016). In professional APCs, Rab11 and Rab22 regulate the presence of intracellular stocks of MHC-I in a compartment resembling the endocytic recycling compartment (ERC), prompting the assumption that these molecules derive from the cell surface (Nair-Gupta et al, 2014; Cebrian et al, 2016). When Rab11 and Rab22 were depleted from murine DCs by shRNA-mediated knockdown, these intracellular MHC-I stocks were depleted and cross-presentation of extracellular antigens was reduced, implying a role for these Rab GTPases in cross-presentation. Significant amounts of MHC-I available for cross-presentation are also found in a presumably recycling compartment in human plasmacytoid DCs (Di Pucchio et al, 2008).

[1]Institut National de la Santé et de la Recherche Médicale, Unité 1151, Paris, France    [2]Université Paris Descartes, Faculté de Médecine, Paris, France    [3]Centre National de la Recherche Scientifique, UMR8253, Paris, France

Correspondence: peter.van-endert@inserm.fr

Arf6 was the first GTPase described to have a role in the endocytic transport of MHC-I (Radhakrishna & Donaldson, 1997). In HeLa cells that overexpress a constitutively active Arf6 mutant, recycling of MHC-I is delayed relative to wild type (WT) cells (Jovanovic et al, 2006) and internalized MHC-I accumulates in endosomal structures coated with F-actin and PIP2 (Donaldson, 2003). However, whether Arf6 is involved in the endocytic trafficking of MHC-I and antigen presentation in professional APCs has not been investigated. Thus, both the extent of MHC-I recycling and the role in it of Arf6, the principal GTPase regulating MHC-I recycling in nonprofessional APCs, remain unclear. Moreover, the potentially distinct roles of folded (peptide–$\beta_2$m HC trimers) and unfolded ($\beta_2$m HC dimers or free HC) MHC-I at the cell surface remain entirely unaddressed.

Having examined MHC-I recycling in Arf6-sufficient and -depleted murine DCs, we report here the surprising finding that, whereas DC MHC-I engages in a fast recycling pathway, internalized folded and unfolded MHC-I do not reach the ERC-like compartment, suggesting that this compartment does not contain recycling MHC-I which, therefore, may not contribute significantly to cross-presentation in murine BM-DCs. Arf6 depletion increases cell surface MHC-I levels, likely by reducing degradation of internalized unfolded MHC-I but does not affect fast recycling. Interestingly, Arf6 depletion affects cross-presentation of immune complexes (ICs) but not that of other antigen forms, suggesting a role of Fc receptor–regulated trafficking of Arf6 vesicles in cross-presentation of ICs.

## Results

### Role of Arf6 in MHC-I cell surface levels

To investigate the role of Arf6 in MHC-I trafficking in BM-DCs, we established a lentivirus-mediated knockdown system that yielded an average knockdown efficiency of 70% (Fig 1A). We first asked whether the absence of Arf6 has an impact on the cell surface levels of the MHC class I allotype H-2K$^b$. We stained transduced BM-DCs using the mAbs B8.28.34 (B8), which recognizes $\beta_2$m-bound K$^b$ HCs (Hämmerling et al, 1982), and AF6-88.5 (AF6), which recognizes peptide-loaded $\beta_2$m-K$^b$ HC complexes (Kuhns & Pease, 1998). Knockdown of Arf6 reproducibly increased surface levels of K$^b$ molecules recognized by either mAbs by 30% (Fig 1B).

Because the antibodies available for K$^b$ do not allow discriminating unfolded from folded MHC-I at the cell surface (Song et al, 1999), we asked whether Arf6 has an impact on the cell surface levels of H-2L$^d$, a class I allotype for which antibodies are available that discriminate unfolded from folded peptide/$\beta_2$m HCs (Hansen et al, 2005). At the steady state, unfolded L$^d$, detected by mAb 64.3.7 (64), was barely detectable on BM-DCs, regardless of the presence or absence of Arf6 (Fig S1A and D–F). However, depletion of Arf6 increased these low levels by 30% as it did those of folded complexes detected by mAb 30.5.7 (30) (Figs 1B and S1A).

We then asked whether this increase in MHC-I at the cell surface correlated with a total increase in cellular MHC-I detectable in fixed and permeabilized cells. Folded, but not unfolded, molecules were about 30% more abundant in the absence of Arf6; although this was not statistically significant because of experimental variation (Figs 1C, and S1B and C). We concluded that the depletion of Arf6 in BM-DCs increases the amount of both folded and unfolded MHC-I of two allotypes at the cell surface.

We next sought to determine whether increased cell surface levels of MHC-I in the absence of Arf6 were because of a lower internalization rate. To do so, we performed a brefeldin A (BFA) decay experiment, which allowed us to follow the removal of cell surface proteins by endocytosis upon disruption of vesicular forward transport (Montealegre et al, 2015). As expected (Machold & Ploegh, 1996), K$^b$ molecules have a long lifetime at the cell surface, with 50% of them present at the cell surface after 16-h treatment with BFA (Fig 1D). Using both AF6 and B8, we could not detect any effect of Arf6 knockdown on the internalization rate of K$^b$ (Fig 1D); the proportional increase in K$^b$ expression in the absence of Arf6 was maintained at all time points (Fig 1E).

Considering reports that H-2L$^d$ molecules have a shorter half-life at the cell surface than other class I allotypes (Mahmutefendić et al, 2007), we reasoned that it might be easier to visualize potential changes in the internalization rate for this allotype than for K$^b$ molecules. Given the very low expression of empty L$^d$ molecules at steady state, we first acid-stripped cells to create a substantial pool of unfolded L$^d$, which at the same time reduced the pool of folded L$^d$ considerably (Fig S1E and F). Acid-stripped and unstripped cells were then subjected to a BFA decay experiment. As expected, unfolded L$^d$ molecules decayed at a faster rate than folded molecules (Fig 1F). However, the relative levels of folded and unfolded L$^d$ in cells lacking Arf6 were 30% higher at all time points during BFA treatment (Fig 1G), suggesting that Arf6 is not involved in regulating the kinetics of MHC-I internalization in BM-DCs, regardless of their conformation.

Because the internalization rate of class I was not affected by the absence of Arf6, we wondered whether an increased forward transport rate might explain the increased MHC-I expression at the cell surface. To address this, we acid-stripped BM-DCs briefly, removing 70% of K$^b$ molecules from the surface, incubated them for various time points at 37°C, and evaluated K$^b$ re-expression by flow cytometry (Fig 1H–J). While there was constantly more class I at the cell surface in the absence of Arf6 even after acid stripping (Fig 1H), the rate of arrival of newly synthesized molecules was slightly slower when Arf6 was depleted (Fig 1I and J). The source of these molecules likely was the secretory pathway because BFA treatment abrogated their arrival to the cell surface (Fig 1H–J). Therefore, an increased forward transport rate from the ER-Golgi to the cell surface was not responsible for the increased amount of MHC-I at the cell surface of Arf6-depleted BM-DCs.

### Role of Arf6 in MHC-I trafficking in the endocytic pathway

To obtain evidence for a potential involvement of Arf6 in MHC-I transport in DCs, we overexpressed Arf6 as a mCherry fusion protein in the H-2$^b$-expressing murine DC line DC 2.4 (Shen et al, 1997; Rong & Shi, 2016) by lentiviral transduction. Consistent with published evidence, Arf6 was present both adjacent to the plasma membrane and in a cytosolic structure (Johnson et al, 2017), surrounded by the cis-Golgi labeled by GM130 (Fig 2). To characterize the internal Arf6$^+$

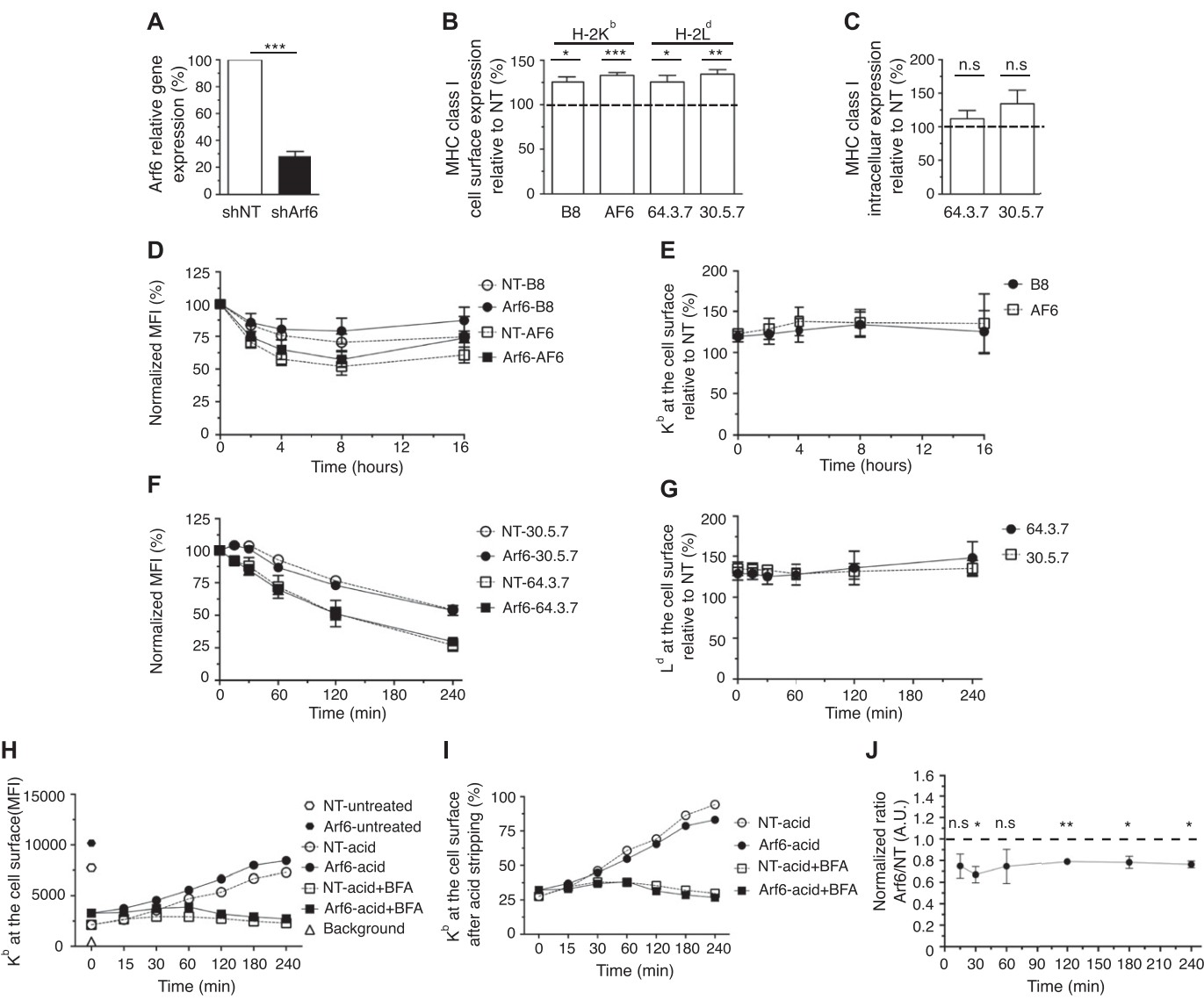

**Figure 1. Effect of Arf6 knockdown on cell surface MHC class I expression.**
BM-DCs were transduced with lentiviruses carrying shRNA sequences targeting Arf6 (shArf6) or a nontargeting (shNT) control. **(A)** Arf6 relative gene expression to shNT was measured by qPCR. Data represent mean and SEM of n = 5 independent experiments. **(B)** Transduced C57BL/6 ($K^b$) or Balb/c ($L^d$) BM-DCs were stained with mAb against $K^b$ (B8 and AF6) or $L^d$ (64 and 30), followed by a fluorescently labeled secondary antibody and a Bv421-conjugated antibody against CD11c. The fluorescence was recorded in the CD11c$^+$ cells. Data represent mean and SEM of n ≥ 6 independent experiments. **(C)** Transduced Balb/c cells were permeabilized and stained with mAbs 64 and 30. Data represent mean and SEM of n = 5 independent experiments. **(D)** Transduced C57BL/6 BM-DCs were incubated with 5 µg/ml BFA; at each time point, cells were stained with the indicated antibodies and analyzed by flow cytometry. **(E)** $K^b$ levels at the cell surface of shArf6-transduced BM-DCs during the BFA treatment were plotted as the percentage of $K^b$ levels of shNT-transduced cells. **(F)** Balb/c BM-DCs were acid-stripped or not and reincubated at 37°C in the presence of 5 µg/ml BFA. Then, acid-treated cells were stained with mAb 64 and untreated cells with mAb 30. **(G)** The $L^d$ levels at the cell surface of shArf6-transduced BM-DCs during the BFA treatment were plotted as the percentage of $L^d$ levels on shNT-transduced cells. Data represent the means and SEM of n = 3 independent experiments. **(H)** Transduced C57BL/6 BM-DCs were acid stripped and reincubated at 37°C in the presence or absence of 5 µg/ml BFA before staining with AF6; a representative experiment is shown. **(I, H)** Data in (H) expressed as a percentage of cell surface values before acid stripping. **(J)** The $K^b$ levels at the cell surface of acid-stripped shArf6-transduced BM-DCs incubated without BFA are plotted as the percentage of $K^b$ levels of shNT-transduced cells. Data represent the means and SEM of n = 3 independent experiments. Data in (A, B, C, J) were evaluated with a one-sample *t* test, under the null hypotheses that the column means of the sample are equal to 100%. Data are significantly different if $P ≤ 0.05$ (*), $P ≤ 0.01$ (**), or $P ≤ 0.001$ (***).

compartment, we costained DC 2.4 cells expressing Arf6-mCherry with markers of the endosomal pathway. Although Arf6 and Rab11 staining did not overlap in peripheral structures, it displayed substantial colocalization in the juxtanuclear region, resulting in an average 40% rate of colocalization at the level of the entire cell (Fig 2A and D). Arf6 also showed lower though significant colocalization

with lysosomal-associated membrzane protein 1 (LAMP-1) in smaller vesicles close to the nucleus likely corresponding to late endosomes/lysosomes (Fig 2B and D), whereas early endosomes antigen 1 (EEA-1$^+$) barely costained with Arf6 (Fig 2C and D).

Given the close proximity of Golgi, lysosome, and recycling compartments to the nucleus, we subjected cells to a short treatment with

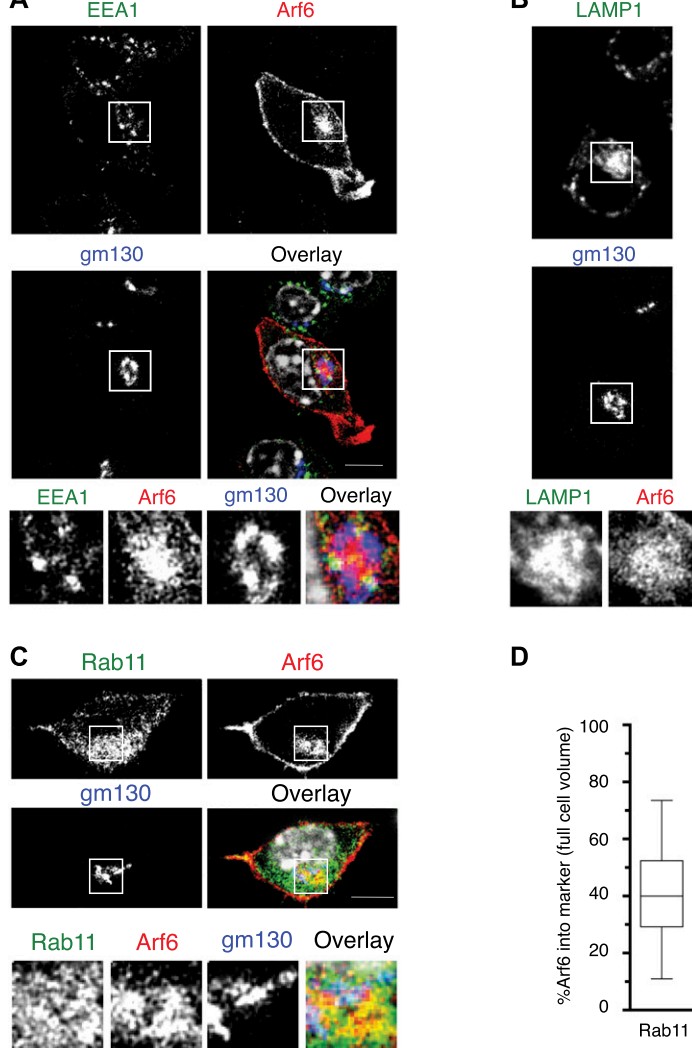

**Figure 2. Intracellular localization of Arf6 in the DC 2.4 cell line.**
**(A, B, C)** Steady state DC 2.4 cells expressing Arf6-mCherry were stained with antibodies against (A) EEA-1, (B) LAMP-1, (C) Rab11. Scale bar: 5 $\mu$m. **(A, B, C, D)** Quantification of (A, B, C). Nuclei are stained with DAPI (grey). Data represent at least 25 cells from n = 3 independent experiments.

nocodazole, inducing microtubule disassembly. This inhibits assembly of the Golgi and distributes the perinuclear compartments to the periphery. As expected, nocodazole treatment fragmented GM130⁺ structures and dispersed lysosomes, Rab11⁺ structures, and vesicles containing MHC-I molecules to the periphery (Fig S2A–D). It similarly dispersed Arf6 vesicles, although some cells still presented a perinuclear Arf6 compartment (Fig S2C and D). Interestingly, both dispersed and remaining perinuclear Arf6 structures still showed significant colocalization with early and late endosomes (Fig S2A and B) but almost completely lacked colocalization with Rab11 and intracellular MHC-I (Fig S2C and D). Collectively, these findings suggest that in DC2.4 cells, Arf6 is distributed through the endocytic pathway and resides mainly at the cell surface and in a juxtanuclear compartment in close vicinity of the Rab11 ERC. Noteworthy, Arf6 is also in contact with putative degradative endosomal compartments.

We next investigated whether Arf6 depletion affected the intracellular distribution of MHC-I that might explain its effect on cell surface levels. Because we failed to detect unfolded K$^b$ molecules after acid stripping by microscopy with mAb B8 (Fig S3A), we used

mAb 64 and 30 that stain L$^d$ in permeabilized Balb/c BM-DCs with high specificity (Fig S1D–F). At the steady state, neither folded nor unfolded L$^d$ colocalized with EEA-1 in transduced Arf6-depleted or control BM-DCs (Fig 3A and B). Next, we examined whether Arf6 affected trafficking in the early endosomal pathway of unfolded L$^d$, internalized upon acid stripping, and of folded L$^d$. Internalized unfolded L$^d$ was readily visualized after 15′ and 30′ in early endosomes, although a substantial pool remained at the cell surface (Fig 3C and E). Consistent with the BFA decay experiments (Fig 1F), internalization of folded L$^d$ was slow, with the first molecules reaching early endosomes after 30′ (Fig 3D and F). Arf6 depletion had no effect on the kinetics of unfolded L$^d$ trafficking to EEA-1 vesicles and resulted in only a minor, statistically not significant delay in the arrival of folded L$^d$ molecules (Fig 3D and F). Thus, internalized unfolded and folded L$^d$ molecules enter early endosomes in an Arf6-independent manner in BM-DCs.

Considering that Arf6 might affect degradation of internalized MHC-I by regulating its routing to degradative compartments, we next monitored arrival of MHC-I to late endosomes or lysosomes. At

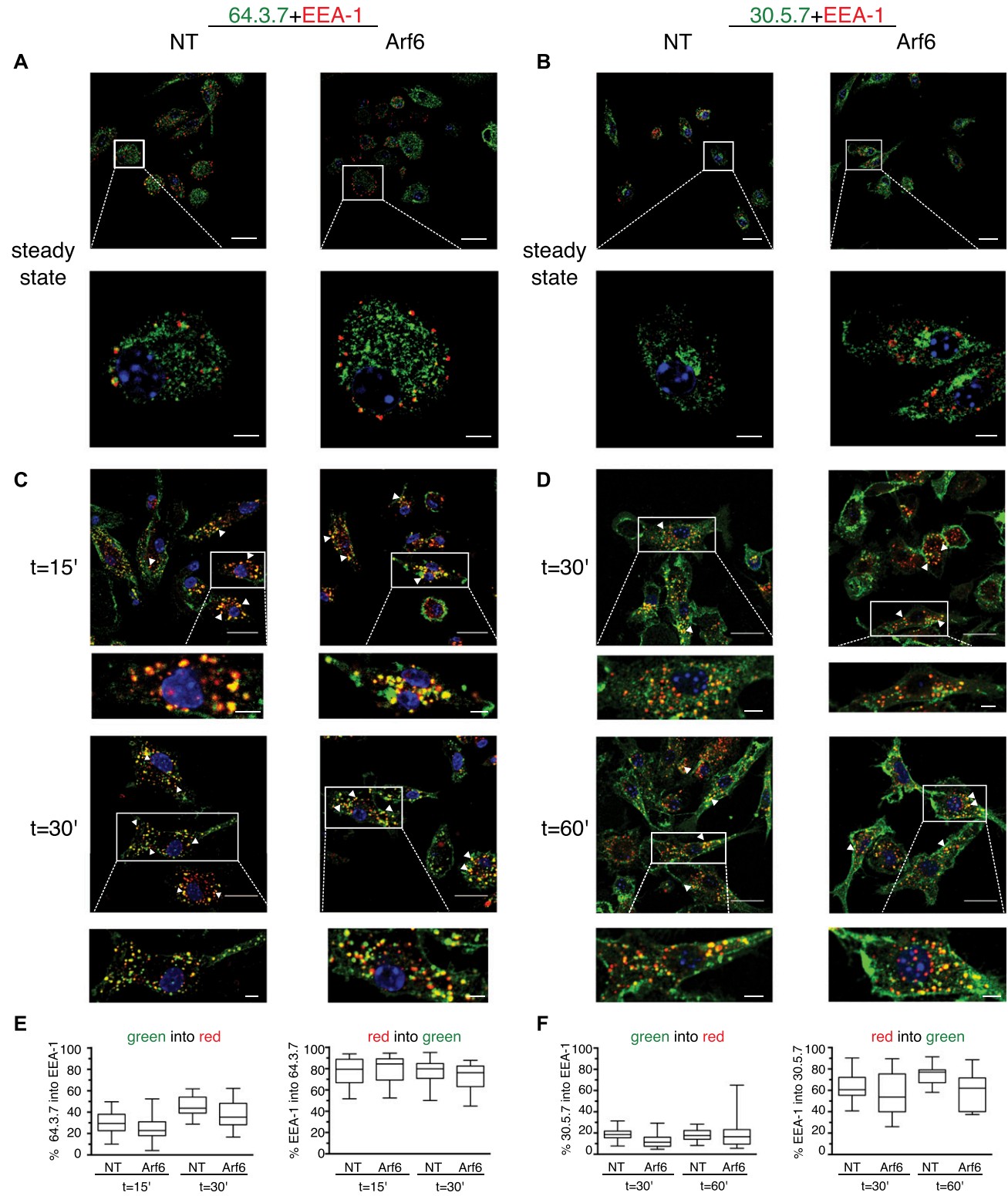

**Figure 3. Folded and unfolded MHC-I enter early endosomes independently of Arf6.**
**(A, B)** Transduced Balb/c BM-DCs were fixed, permeabilized, and stained with a combination of antibodies: EEA-1 (red) and 64 (green, (A)) or 30 (green, (B)). **(A, C, D)** Cells were preincubated with 64 (after acid stripping) (C) or 30 (D) on ice water 30 min. After washing excess antibodies, cells were incubated for the indicated time points at 37°C, and then treated as in (A). **(C, D, E, F)** Quantification of (C) and (D). Nuclei are stained with DAPI (blue). Data represent at least 15 cells per condition from two independent experiments. Nuclei are stained with DAPI (blue). Scale bars = 20 μm in upper panels, and 5 μm in lower panels. NT, nontargeting shRNA.

the steady state, some unfolded $L^d$ colocalized with LAMP-1, although most 64-reactive molecules resided in perinuclear compartments, presumably the ER (Fig 4A). Folded $L^d$ was also detected in the latter compartments but absent from LAMP-1$^+$ vesicles (Fig 4B). When internalization of cell surface molecules was monitored, unfolded and folded $L^d$ molecules remained detectable after 1 and 2 h, respectively, but did not colocalize with LAMP-1 at these time points (not shown). However, unfolded $L^d$ molecules were no longer visible after 2 h and folded $L^d$ molecules after 4 h of internalization (Fig 4C and D). Considering that this was likely because of their degradation, we treated cells with concanamycin B (ConB), a specific inhibitor of V-ATPase that blocks endosomal acidification. This resulted in the appearance of some unfolded $L^d$ close to LAMP-1$^+$ compartments, although most was still in EEA-1$^+$ vesicles after 2 h (Fig 4E). Interestingly, depletion of Arf6 from ConB-treated DCs increased the amount of unfolded $L^d$ detected at this time point significantly (Fig 4E and G). ConB incubation also resulted in the appearance of significant amounts of folded $L^d$ that colocalized partially with LAMP-1, however, without an effect of Arf6 depletion (Fig 4F and H). Taken together, these results suggest that internalized unfolded and folded $L^d$ molecules travel to LAMP-1 compartments where they are degraded in a pH-sensitive manner, and that Arf6 might promote degradation of unfolded $L^d$ through this pathway, consistent with its partial colocalization with LAMP-1 (Fig 2B and D).

## The source of MHC-I in DC Rab11+ vesicles

Arf6 has been reported to mediate recycling of HLA class I in HeLa cells (Radhakrishna & Donaldson, 1997; Jovanovic et al, 2006). Given that significant amounts of MHC-I have been reported to reside in a Rab11$^+$ compartment in BM-DCs (Nair-Gupta et al, 2014), it is commonly assumed that internalized MHC-I molecules to the ERC in BM-DCs from which they can be recruited to phagosomes for cross-presentation. To test this directly, we first analyzed colocalization of $L^d$ with Rab11a at the steady state (Fig 5A and B). Rab11 localized to a rounded structure in the perinuclear area, resembling the ERC (Goldenring, 2015), and to vesicles emanating from the perinuclear area. The absence of Arf6 did not alter this staining pattern (Fig 5A and B). Unfolded $L^d$ molecules resided mostly outside the ERC, although a minor population was found within it (Fig 5A). Conversely, folded $L^d$ molecules colocalized strongly with the ERC-like structure, in agreement with a previous report (Nair-Gupta et al, 2014) (Fig 5B). Arf6 depletion had no detectable effect on the distribution of folded or unfolded MHC-I to Rab11$^+$ structures at the steady state.

Next, we examined the hypothesis, suggested by the literature, that $L^d$ molecules residing in the ERC-like structure were derived by internalization from the cell surface. However, to our surprise, although internalization of both folded and unfolded $L^d$ molecules was readily detectable (Figs 3, 4, and 5C and D), we failed to observe significant amounts of $L^d$ reaching the ERC during a chase period of up to 1 h, a time reported for the arrival of MHC-I to the ERC in nonprofessional APCs (Jovanovic et al, 2006; Zagorac et al, 2012) (Fig 5C and D). The same was true for later time points up to the disappearance of internalized $L^d$ (not shown), and whether Arf6 was depleted or not. Thus, it was unlikely that $L^d$ molecules residing in

the ERC or in peripheral Rab11$^+$ vesicles were derived from the cell surface in any significant proportion.

To confirm this surprising finding, we next monitored trafficking of internalized H-2K$^b$ molecules in Rab11-counterstained Arf6-mCherry–transfected DC2.4 cells. First, using "exon 8," a conformation-independent antiserum recognizing the cytosolic tail of H-2K$^b$ and H-2D$^b$ (Day et al, 1995), we observed a strong MHC-I presence in a perinuclear compartment reminiscent of the one detected in BM-DCs with mAb 30 (Fig 5B). Remarkably, class I staining in this perinuclear area was intermingled and showed overlap with Arf6-mCherry fluorescence (Fig 6A). As discussed above (Fig S2D), nocodazole treatment dispersed intracellular MHC-I molecules to peripheral cytosolic structures not overlapping with Arf6 or nascent cis-Golgi structures. Thus, MHC-I in DC 2.4 cells, as in BM-DCs, reside in a perinuclear, nocodazole sensitive, Arf6$^+$ compartment (Fig 2A).

To address whether internalized MHC-I reach the Arf6 internal compartment, we monitored the uptake of mAb-labeled cell surface molecules. Before internalization, folded K$^b$-peptide complexes (detected with mAb AF6) colocalized at the cell surface with Arf6, but not with the internal Arf6 structure (Fig 6B and C). Because of the long half-life of AF6-reactive K$^b$ (Fig 1D), little K$^b$ was internalized after 20 or 60 min of incubation at 37°C. However, of these internalized K$^b$ molecules, very few, if any, reached the Rab11$^+$/Arf6$^+$ compartment (Fig 6B and C). Similarly, the few K$^b$ molecules internalized after 2 h in BM-DCs localized to small endocytic vesicles rather than to the Rab11$^+$ ERC (Fig S3B). Thus, different MHC class I allotypes and conformers fail to travel upon internalization to the Rab11$^+$Arf6$^+$ compartment in BM-DCs and in the DC2.4 line.

## Fast recycling of MHC-I in BM-DCs

Considering the unexpected finding that internalized $L^d$ and K$^b$ molecules traveled to early endosomes and lysosomes, but not to the Rab11$^+$Arf6$^+$ ERC, we wondered whether MHC-I recycle at all in BM-DCs. To test this directly, we chose to adapt a recycling assay described by Ma et al, in which MHC-I are allowed to internalize in the presence of primaquine (Ma et al, 2016). Given that primaquine is known to inhibit receptor recycling including that of MHC-I in nonprofessional APCs, it is expected that primaquine incubation results in MHC-I accumulation in swollen endosomes, which then may be released upon drug removal. However, although primaquine incubation rapidly reduces the cell surface amounts of a typical recycling receptor, for example, the transferrin receptor, it did not affect the cell surface density of K$^b$ molecules after 15 or 30 min (Fig 7A). Nevertheless, when we removed the remaining AF6-bound K$^b$ molecules after 30 min of internalization in the presence of primaquine, followed by another incubation for up to 60 min to allow for export of accumulated MHC-I, 65% of the internalized K$^b$ molecules reappeared at the cell surface within 5–15 min versus 15–20% in the absence of the drug (Figs 7B and S4). The latter result suggests that fully conformed K$^b$ molecules may recycle via a fast pathway in BM-DCs. We also asked whether Arf6 might regulate such a fast recycling pathway. However, Arf6 depletion had no significant effect on fast K$^b$ recycling, as observed after an hour of K$^b$ internalization (Fig 7C–D).

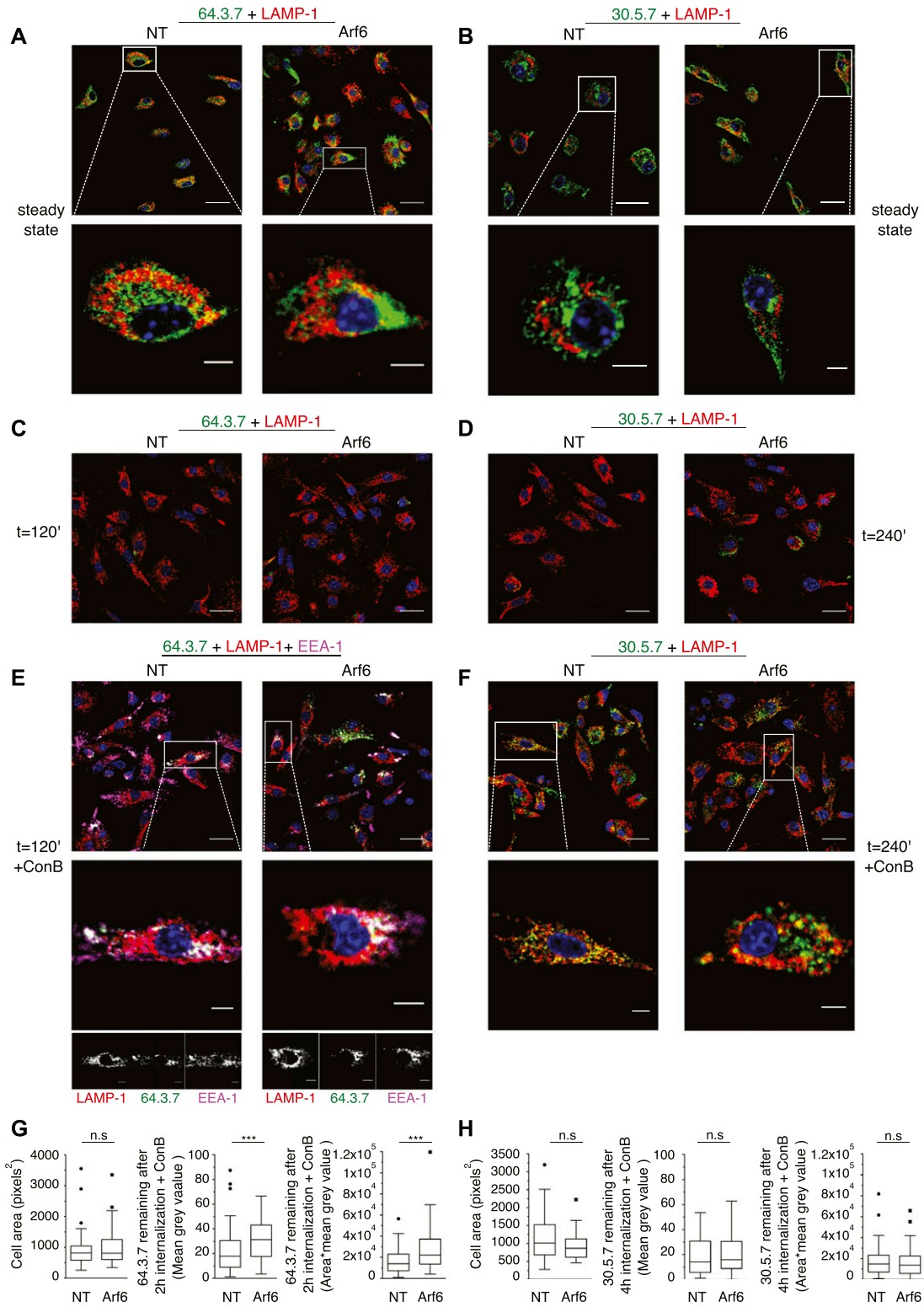

**Figure 4. Delayed disappearance of unfolded MHC-I in Arf6-depleted cells.**
**(A, B)** Transduced Balb/c BM-DCs cells were fixed, permeabilized, and stained with antibodies against LAMP1 (red) and 64 (green, (A)) or 30 (green, (B)). **(A, C, D)** Cells were preincubated with 64 (after acid stripping) (C) or 30 (D) on ice water 30 min, washed, incubated for the indicated duration at 37°C, and treated as in (A). **(E)** Cells were preincubated with 20 nM ConB, acid-stripped, pulsed with 64 on ice water 30 min, reincubated for 2 h at 37°C with ConB, fixed, and stained with antibodies against LAMP1 (red), EEA-1 (magenta), and a secondary antibody against 64 (green). **(E, F)** Cells were treated as in (E) but without acid stripping and stained with mAb 30 (green). **(E, G)** Quantification of (E). **(F, H)** Quantification of (F). The area (left panel) and mean grey values (center panel) of each cell were quantified as described in material and

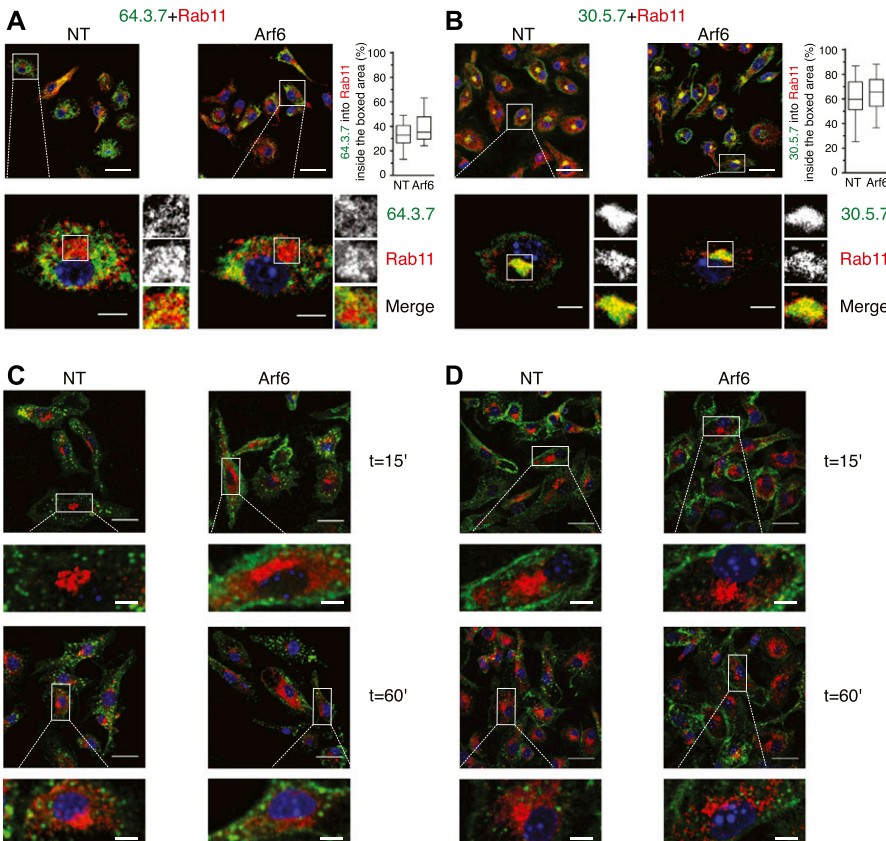

**Figure 5. Folded MHC-I residing in the ERC are not derived from the cell surface.**
**(A, B)** Transduced Balb/c BM-DCs cells were fixed, permeabilized, and stained with antibodies against Rab11a (red) and with 64 (green, (A)) or 30 (green, (B)). **(A, C, D)** Cells were pulsed with 64 (after acid stripping, (C)) or 30 (D) on ice water, incubated for the indicated time points at 37°C, and treated as in (A). Nuclei are stained with DAPI (blue). Data represent at least 27 cells per condition from n ≥ 3 independent experiments. Scale bars = 20 μm in upper panels, and 5 μm in lower panels. NT, nontargeting shRNA.

## Impact of Arf6 in cross-presentation

Although our results so far did not suggest a role of Arf6 in MHC-I recycling, we reasoned that Arf6 colocalization with the ERC, as well as our observation of increased cell surface MHC-I levels in its absence, could be related to a role in cross-presentation. To evaluate this, we first studied presentation of the ovalbumin (OVA) peptide SIINFEKL (SL8) to naive OT-I T cells recognizing this peptide in the context of $K^b$. We found no difference in the ability of Arf6-depleted BM-DCs to stimulate OT-I cells relative to control cells (Figs 8A and S5A). Therefore, the increase in $K^b$ levels at the cell surface caused by Arf6 depletion was not sufficient for generating a stronger CD8$^+$ T cell priming.

We then sought to determine whether Arf6 plays a role in cross-presentation of different antigen forms. We first tested a synthetic long peptide (GS-20) but found no effect of Arf6 knockdown on cross-presentation (Figs 8B and S5B), reminiscent of a recent report that shows that the cross-presentation of a long peptide by human moDCs is independent of Rab11 (Ma et al, 2019). The same was true for cross-presentation of yeast cells decorated with full length OVA and taken up by phagocytosis (Figs 8C and S5C). Next, we tested ICs formed by OVA and specific antibodies and internalized via the Fc$_\gamma$

receptors. Strikingly, we observed a 50% reduction in OT-I T cell stimulation by Arf6-depleted BM-DCs relative to control cells (Figs 8D and S5D). To determine whether the same effect applied to antigens internalized by other receptors, we used antibodies to target the OVA-containing fusion protein P3UO (Kratzer et al, 2010) to the mannose receptor (Zehner & Burgdorf, 2013) or to CD11c and evaluated cross-presentation as above. Surprisingly, Arf6 depletion had no effect on cross-presentation of antigens targeted to these receptors (Figs 8E and F, and S5E and F). Thus, Arf6 is specifically required for cross-presentation of ICs taken up by Fc$_\gamma$ receptors of BM-DCs, but dispensable for cross-presentation of other antigens including long peptides, phagocytosed, and receptor-targeted antigens.

To explore the potential mechanism that Arf6 uses to control cross-presentation of ICs, we studied uptake and intracellular routing of ICs. The cell surface levels of CD16, CD32, and CD64 were unaltered by the depletion of Arf6 (Fig 9A). Moreover, the binding of ICs to Arf6 depleted cells was not altered (Fig 9B). In RAW.264 macrophages expressing the dominant negative mutant Arf6 S21N, the internalization of IgG-opsonized sheep red blood cells has been described to be compromised (Niedergang et al, 2003). However, ICs visualized with primary and secondary antibodies, were readily

methods, and the product *area × mean grey value* was expressed as the integrated density. Data in (E) and (G) are from at least 62 cells from three independent experiments. Data in (F) and (H) are from at least 42 cells from three independent experiments. Data were evaluated with an unpaired nonparametric Mann–Whitney test, $P ≤ 0.05$ (*), $P ≤ 0.01$ (**), or $P ≤ 0.001$ (***). Nuclei are stained with DAPI (blue). Scale bars = 20 μm in upper panels, and 5 μm in lower panels. NT, nontargeting shRNA.

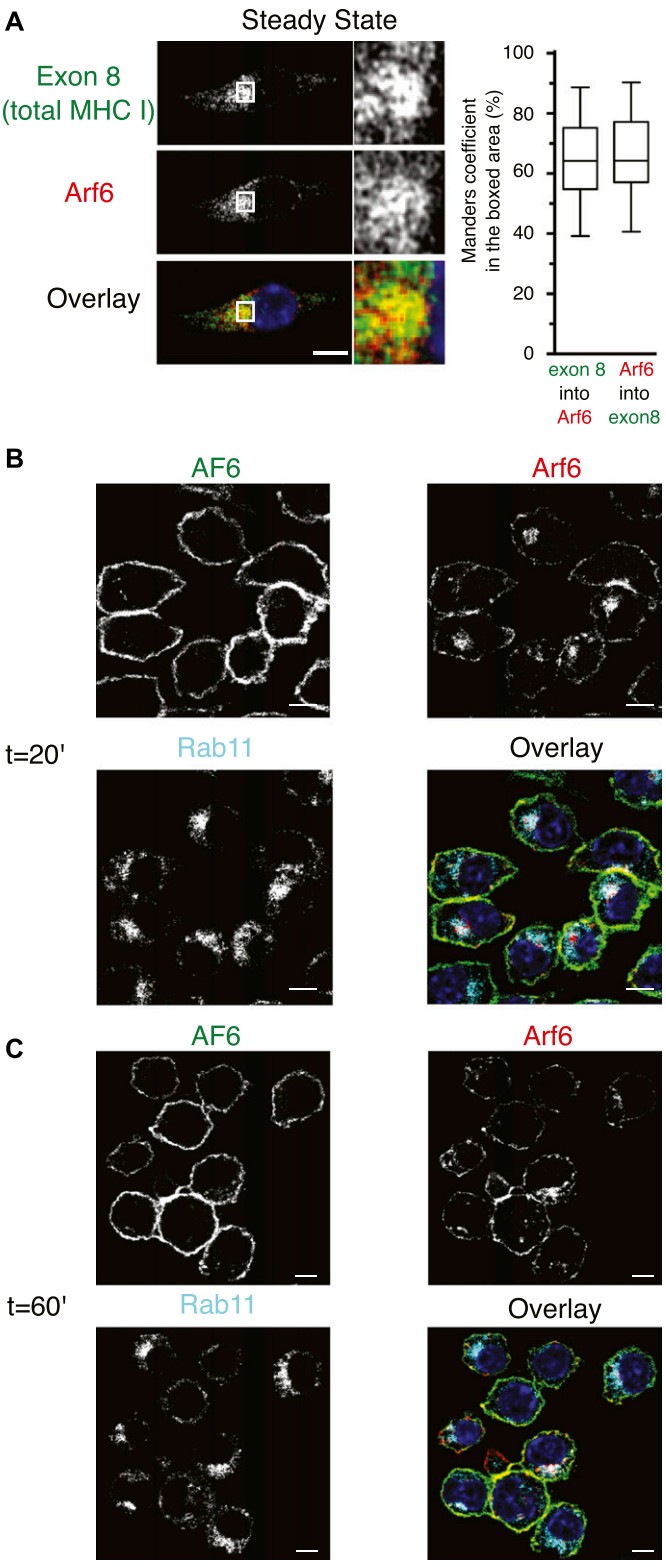

**Figure 6.  MHC-I colocalizing with Arf6 is not derived from the cell surface.**
**(A)** DC 2.4 cells expressing Arf6-mCherry were fixed, permeabilized, and stained with the antibody exon 8. Quantification represents 30 cells from three independent experiments. **(B, C)** DC 2.4 cells expressing Arf6-mCherry were pulsed on ice with AF6, incubated for 20 (B), or 60 min (C), at 37°C to allow for internalization, fixed, permeabilized, and stained for Rab11 and AF6

internalized in the presence or absence of Arf6, reaching early endosomes after 10' in similar proportions (Fig 9C and E). As long as ICs remained visibly intracellular, they did not appear in LAMP-1[+] lysosomes (Fig 9D). The timing of IC disappearance at 1 h also was not affected by Arf6 depletion.

We then wondered whether enhanced antigen degradation could explain compromised cross-presentation of ICs in Arf6-depleted cells. We used a FACS-based assay measuring degradation of self-quenching Alexa Fluor 647–conjugated OVA that emits an increasing fluorescent signal upon degradation. Soluble OVA-647 was degraded at a similar rate in BM-DCs in the presence or absence of Arf6 (Figs 9F and S6A), reaching plateau after 2 h. Antibody binding to OVA-647 at 4°C increased fluorescence, presumably because of steric effects (Fig S6B and C). Upon incubation at 4°C, OVA-647 ICs were steadily degraded; however, in this case, degradation was increased in the absence of Arf6 (Figs 9G and S6B).

Finally, we incubated DC2.4 cells expressing Arf6-mCherry with complexes of self-quenching fluorogenic DQ-OVA and OVA antibodies (Fig S6C) to perform live-cell imaging experiments, asking whether ICs encounter Arf6 during their intracellular transport. Interestingly, we observed that upon addition of ICs to DC2.4 cells, the internal Arf6 structure moved from the perinuclear area to the cell surface, seemingly to encounter the incoming antigen (Videos 1–5 and Fig 9I). In contrast, upon addition of soluble DQ-OVA to the cells, smaller vesicles containing the incoming antigen typically traveled to the Arf6 compartment which itself did not move (Videos 6 and 7, and Fig 9H). Taken together, our data suggest that although Arf6 is not required for cell surface binding, uptake, and routing to early endosomes of IC, the Arf6 compartment comes into close contact with both soluble, likely pinocytosed antigen and ICs. The distinct intracellular trafficking of these antigen forms suggests a role of signaling by the Fc receptor in mobilization of the Arf6 compartment.

## Discussion

In this study, we report several unexpected findings. Given ample evidence for efficient MHC-I recycling in nonprofessional APCs and the ready detection of MHC-I molecules in a Rab11[+] compartment, we expected to find routing of internalization MHC-I into the ERC of BM-DCs but failed to do so. Our results are not in conflict with reports that TLR4 signaling triggers delivery of MHC-I from the ERC to phagosomes (Nair-Gupta et al, 2014), or that TLR signaling together with MHC-I ligation triggers tubulation of the ERC toward the surface in human monocyte–derived DCs (Compeer et al, 2014). They also do not contradict reports that recruitment of MHC-I molecules stored in an ERC-like compartment in DCs is important for cross-presentation. However, they suggest that the endosomal MHC-I molecules delivered through these mechanisms are not derived from the cell surface and call for further investigation to identify their origin. Because we tracked internalized

(secondary antibody only). Nuclei are stained with DAPI (blue). Scale bars 5 = μm.

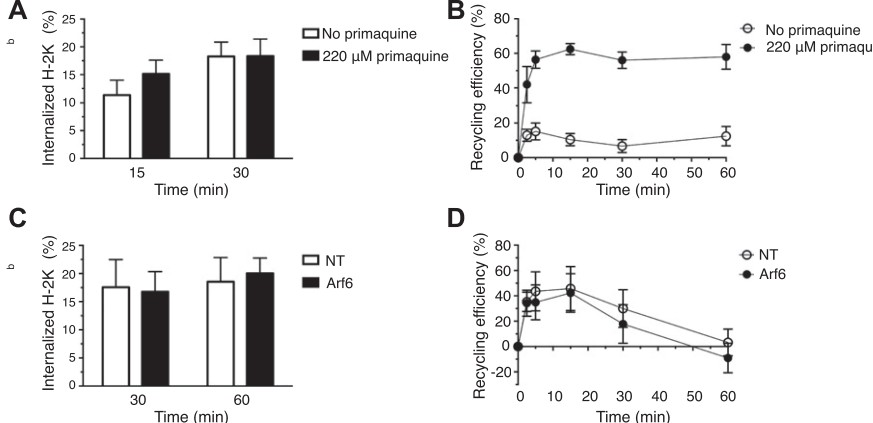

**Figure 7. MHC-I recycles through a fast Arf6-independent pathway in BM-DCs.**
**(A)** C57BL/6 BM-DCs were pulsed with AF6 for 30 min on ice and shifted to 37°C for 30 min in the presence or absence of primaquine to allow for internalization and stained for AF6 using a secondary antibody.
**(A, B)** BM-DCs were treated as in (A) including the internalization step for 30′, acid stripped, reincubated at 37°C in the absence of primaquine to allow for recycling, and finally stained with a secondary antibody for analysis by flow cytometry.
**(C)** Transduced C57BL/6 BM-DCs were pulsed with AF6 as in (A) and then allowed to internalize AF6-bound H-2K$^b$ for 60 min in the presence of primaquine. **(B, D)** Recycling of H-2K$^b$ was analyzed as in (B), following an internalization step of 60 min. **(A, B, C, D)** Data represent means and SEM of n = 3 (A, B) and n = 4 (C, D) independent experiments. NT, nontargeting shRNA.

MHC-I by means of mAb bound on the cell surface, it could be argued that intracellular MHC-I trafficking might become undetectable upon pH-dependent dissociation of MHC-I/mAb complexes and degradation of mAb. However, the fact that we could readily detect MHC-I in early endosomes and even in late endosomes in the presence of ConB but not in the ERC, although pH in the former (6.0–6.5) is lower than in the latter (6.5–7.0) (Modi et al, 2013), argues against this hypothesis. We also cannot rule out that antibody binding affects intracellular trafficking of MHC-I molecules, a possibility that could only be addressed by covalent labeling of cell surface MHC-I molecules, for example, by biotinylation. We attempted to adapt a published FACS assay to monitor the fate of peptide-bound MHC-I molecules (Dumont et al, 2017), but failed because of the low signal to noise ratio obtained (not shown).

The absence of an effect of Arf6 on cross-presentation was equally unexpected given that knockdown of Rab22, acting in an Arf6-dependent recycling pathway in nonprofessional APCs, compromises cross-presentation (Cebrian et al, 2016). However, in nonprofessional APCs, Rab22 is involved in the formation of tubular endosomes emanating from the ERC, as opposed to Arf6 that

mediates clathrin-independent MHC-I internalization (Weigert et al, 2004). Therefore, the effect of Rab22 in cross-presentation may reflect a role in recruiting MHC-I from the ERC but not in routing internalized MHC-I to it. In other words, we propose that the role of the small GTPases Rab11 and Rab22 to mediate MHC-I exit from the ERC in nonprofessional APCs is conserved in DCs, whereas the role of Arf6 in internalization, and with it the mechanism for loading the ERC with MHC-I, is not.

Our observations concerning the intracellular localization of MHC-I at the steady state are consistent with published data. Unfolded molecules are found in the ER, presumably just after their synthesis, and to a minor extent in late endosomes/lysosomes, probably awaiting degradation, although feeding into a putative novel recycling pathway (Mahmutefendić et al, 2017) is not ruled out by our data. Folded molecules are mainly found in the ERC-like structure which is devoid of unfolded MHC-I. Upon internalization, folded and unfolded MHC-I travel through EEA-1$^+$ early endosomes to late endosomes/lysosomes, presumably for degradation by pH-dependent proteases as revealed in the ConB experiment.

Experiments using primaquine suggested that MHC-I recycling is not absent from DCs but follows a somewhat unexpected fast

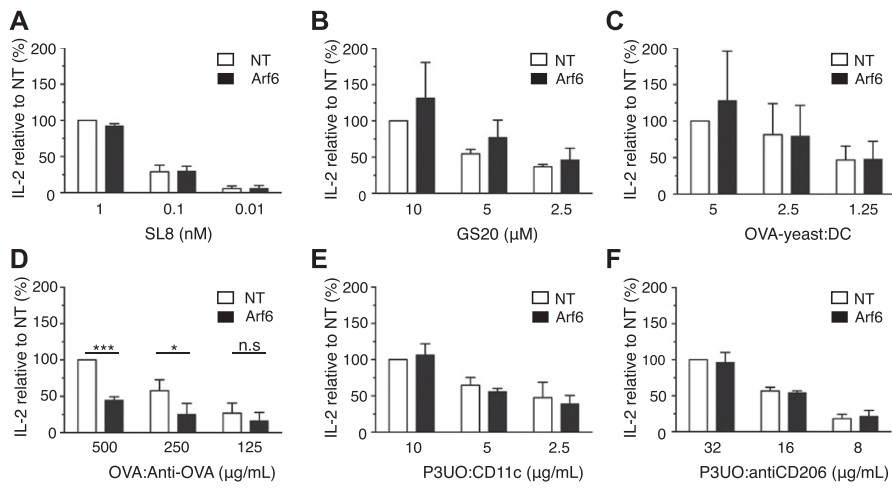

**Figure 8. Effect of Arf6 depletion on cross-presentation of various antigen forms.**
**(A)** Transduced C57BL/6 BM-DCs were pulsed with various amounts of SL8 peptide, fixed, and cocultured with OT-I T cells. The next day, the supernatant was recovered and probed for IL-2 by ELISA. **(B, C, D, E, F)** Cross-presentation to OT-I cells of peptide GS-20 (B), of irradiated yeast cells expressing full length OVA at the cell surface (C), of OVA-anti-OVA IC (D), and of the OVA fusion protein P3UO targeted to CD11c (E), or CD206 (F).
**(A, B, C, D, E, F)** Data represent the means and SD of n = 3 independent experiments (A, B, C, E, F) and n = 4 (D). Data were evaluated with a one-sample *t* test, under the null hypotheses that the column means of the sample are equal to 100%. Data are significantly different if $P \leq 0.05$ (*), $P \leq 0.01$ (**), or $P \leq 0.001$ (***). NT, nontargeting shRNA.

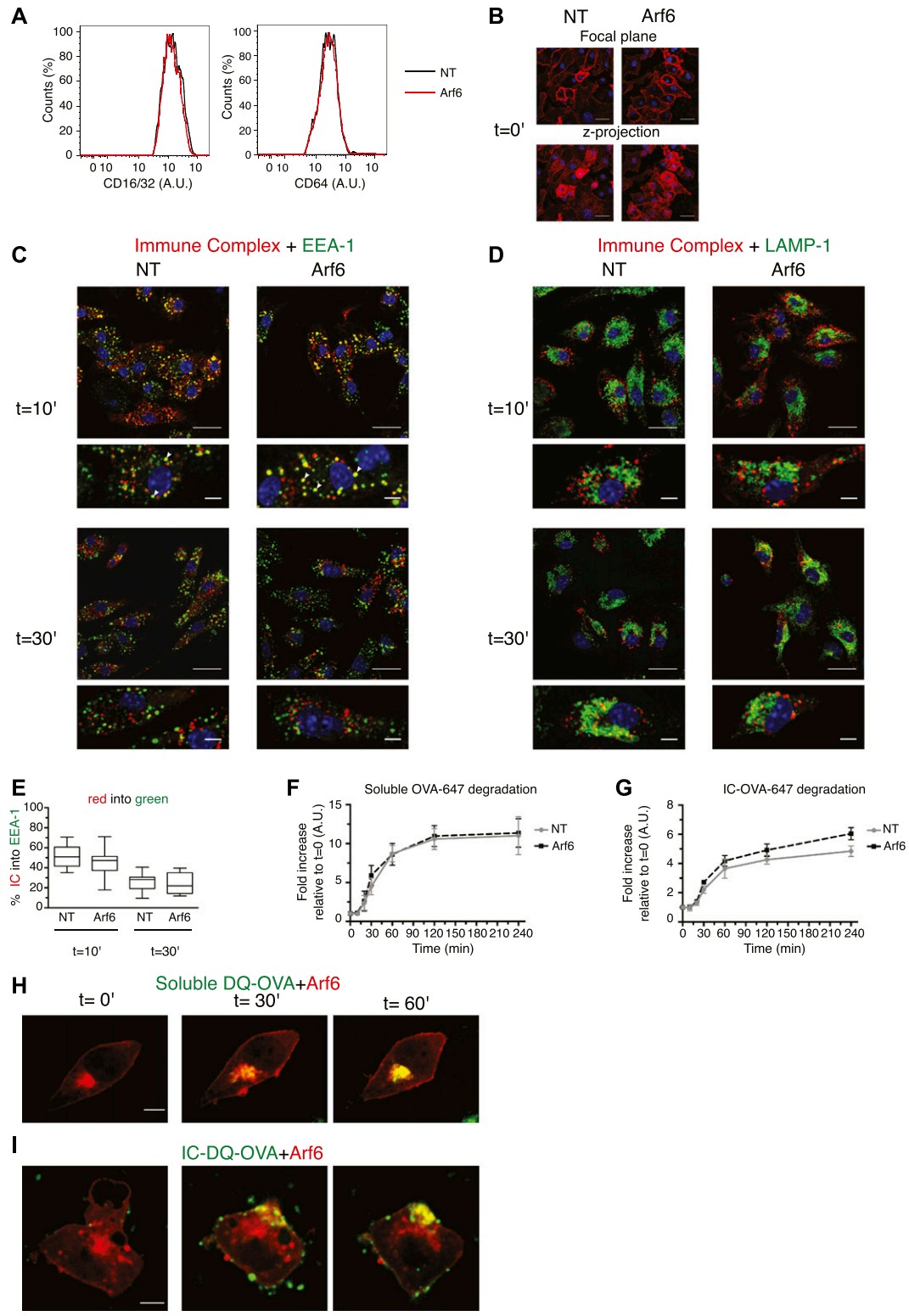

**Figure 9. Intracellular routing and degradation of ICs and soluble OVA.**
**(A)** FACS analysis of FcγRI/II cell surface expression in transduced BM-DCs. **(B)** Arf6-depleted and control BM-DCs were analyzed for IC binding to the surface by pulsing at 4°C with OVA–anti-OVA complexes. Upper panels: focal plane; Lower panels: cumulative z-projection. Scale bar: 20 μm. **(C, D)** After binding at 4°C, ICs were allowed to internalize at 37°C for 10 or 30 min and cells were stained for EEA-1 (C), and LAMP-1 (D), and a secondary antibody against the OVA antibody. Scale bars, upper panels: 20 μm; lower panels 5 μm. **(C, E)**, Quantification of (C). Data represent at least 15 cells from three independent experiments. Nuclei are stained with DAPI (blue). **(F)**, Transduced BM-DCs were pulsed with soluble OVA-647 on ice and incubated at 37°C for the indicated time points, before recording OVA fluorescence by flow cytometry. Data

pathway. The rate of MHC-I internalization was relatively modest, with internalization of only 15% in 15 min, compared, e.g., with internalization of 85% of transferrin within 6 min in the presence of primaquine (Stoorvogel et al, 1987). Fast MHC-I recycling has been described for a Rab35 and clathrin-dependent pathway in COS cells (Allaire et al, 2010). However, the kinetics of recycling through this pathway (almost complete export of accumulated intracellular MHC-I in <5 min) is not consistent with the slow Rab11-dependent recycling pathway implicating the ERC described in the literature, and unlikely to be sufficient for peptide loading of recycling MHC-I.

Although we were unable to detect any effect of Arf6 depletion on fast MHC-I recycling and for MHC-I accumulation in the ERC, Arf6 knockdown reproducibly increased cell surface MHC-I levels by 30–50%, most likely by reducing degradation of internalized un-folded MHC-I. Contrary to a previous report (Nair-Gupta et al, 2014), we find substantial colocalization of Arf6 with Rab11, however, without a structural role, because the juxtanuclear Rab11$^+$ MHC-I reservoir is unaltered in its absence. However, nocodazole treat-ment suggested that the Arf6 structures may reside in close proximity to, rather than be identical to, the perinuclear Rab11+ and also MHC-I–containing perinuclear structures. Experiments using super-resolution microscopy should be helpful to identify the precise nature and dynamics of the different perinuclear com-partments in DCs.

Not having tested MHC-I trafficking upon signaling by TLR4 concomitant with phagocytosis, we do not know whether Arf6 plays any role in the reported LPS-induced mobilization of MHC-I to phagosomes. We noted a striking and exclusive effect of Arf6 de-pletion on cross-presentation of ICs. Unlike cross-presentation of soluble OVA, long peptides or CD206/CD11c-targeted OVA, cross-presentation of ICs is associated with signaling by Fcγ receptors which, for example, can mobilize ill-defined factors to individual phagosomes that promote cross-presentation (Hoffmann et al, 2012). Reminiscent of our findings, signaling by FcγR has recently been described to activate Arf6 and recruit it to phagosomes, where it stimulates phospholipase D to produce phosphatidic acid re-quired to complete phagocytosis (Tanguy et al, 2019). The encounter and engulfment of ICs by Arf6 structures visualized in live video microscopy (Fig 9I and Videos 1–5) may result from similar FcγR-triggered signaling events possibly involving phospholipase D that are required for efficient cross-presentation of ICs. The absence of this phenomenon in cells engulfing pinocytosed OVA is consistent with the hypothesis that Fc receptor–triggered signaling acting on the Arf6 compartment and its trafficking are responsible for com-promised cross-presentation in Arf6-depleted DCs.

If MHC-I in the ERC is not derived by internalization from the surface, what can be its origin? Several recent reports propose pathways that divert newly synthesized MHC-I to endosomal/phagosomal compartments for cross-presentation. Gatti and col-leagues demonstrated that MHC-I ubiquitination by the ligase MARCH9 diverts MHC-I from the TGN to Syntaxin6$^+$ endosomes, an event promoting cross-presentation (De Angelis Rigotti et al, 2017).

Van den Eynde et al suggested that cross-presented long synthetic peptides can be loaded on newly synthesized MHC-I, accessing endosomes through a nonconventional secretory pathway (Ma et al, 2016). Together with these reports, our observations challenge current concepts about the origin of cross-presenting MHC-I and call for new efforts to track MHC-I trafficking in DCs.

# Materials and Methods

## Reagents and antibodies

Unless indicated otherwise, laboratory reagents were purchased from Sigma-Aldrich. U-bottomed 96-well plates (Cat. No 650185) for lentiviral transduction were purchased from Dutscher (France). OVA antibodies (whole antiserum produced in rabbit, ref. C6534) were purchased from Sigma-Aldrich; the IgG fraction was further purified by affinity chromatography. MAb mouse anti H-2K$^b$ clone AF6-88.5-biotin, purified mouse anti-mouse CD64 (clone X54-5/7.1), purified rat anti-mouse CD16/32 (clone 93), and streptavidin-conjugated APC-Cy7 were from BioLegend. MAb rat anti-mouse CD206 (clone MR5D3) was from Bio-Rad. MAb Armenian hamster anti-mouse CD11c (clone N418) conjugated to BV421, purified rat anti-mouse IL-2 (clone JES6-1A12), biotinylated rat anti-mouse IL-2 (clone JES6-5H4), rat anti-mouse CD107a (LAMP-1, clone 1D4B), and 7-amino-actinomycin D (7-AAD) live/dead staining solution were from BD Bio-sciences. Polyclonal goat anti-mouse EEA-1 (N19) was from Santa Cruz. Polyclonal rabbit anti-mouse Rab11a (#2413) was from Cell Signaling Technology. Polyclonal donkey anti-mouse coupled to Alexa Fluor 488, donkey anti-rat coupled to Alexa Fluor 594, donkey anti-goat coupled to Alexa Fluor 594, goat anti rat coupled to Alexa 647, DQ-OVA, and Ov-albumin Alexa-Fluor 647 were purchased from Life Technologies. The N418 hybridoma–secreting Armenian hamster anti-mouse CD11c was purchased from the American Type Culture Collection. Hybridomas 30.5.7 and 64.3.7 were a gift of J Reimann, University of Ulm. MAb mouse anti H-2K$^b$ clone B8.24.3 ascites was a gift of F Lemonnier, Institut Pasteur. Anti-exon-8 antibody was a gift from Jack Bennink (NIH). 3,3′,5,5′-tetramethylbenzidine (TMB) substrate reagent set (BD OptEIA) was purchased from BD Biosciences. OVA purified from egg white (dialyzed, lyophilized powder) was purchased from Worthington Bio-chemical Corporation. Peptides SL8 (>80% purity) and GS20 (GLEQLE-SIINFEKLTEWTSS, 85.8% purity) were custom-synthesized by Schafer-N.

## Mice and cells

C57BL/6 and RAG-1 deficient OT-I T cell receptor–transgenic mice were maintained in our animal facility. BALB/c mice were pur-chased from Janvier. BM-DCs were produced as previously de-scribed (Lawand et al, 2016) using 3% J558 hybridoma supernatant as a source of GM-CSF. BM-DCs were used for experiments on day 7 of differentiation. DC 2.4 cells (Shen et al, 1997) were cultured in

---

represent mean and SEM from n = 3 independent experiments. **(F, G)** As in (F), with the IC formed between an OVA antibody and OVA-647. **(H)** DC 2.4 cells expressing Arf6-mCherry were fed soluble DQ-OVA at 37°C, and the uptake was followed by live-cell imaging for 1 h. Scale bar: 5 *µ*m. **(H, I)** As in (H), with IC containing DQ-OVA. Data in (H) and (I) represent n = 2 independent experiments, with 10 cells per condition. NT, nontargeting shRNA.

Roswell Park Memorial Institute (RPMI)-1640 (medium) with 10% heat inactivated FBS (Eurobio Abcys), 100 U/ml penicillin, 100 µg/ml streptomycin, 2 mM L-glutamine, 50 µM β-mercaptoethanol, without GM-CSF, and divided every other day.

## Plasmids, lentivirus production, and transduction

A pLK0.1 plasmid carrying a shRNA sequence specific for Arf6 was purchased from Sigma-Aldrich (shArf6; MISSION clone NM_007481.3-769s21c1, TRCN0000324934, sequence 5′-CCGGGCATTACTACA-CCGGGACCCACTCGAGTGGGTCCCGGTGTAGTAATGCTTTTTG-3′). Nontargeting shRNA (shNT; SHC002H; Sigma-Aldrich) was used as a control. Both plasmids contained a puromycin resistance gene. Plasmid DNA was extracted and purified with Nucleobond Xtra Maxi EF kit (Macherey-Nagel) and used to produce lentiviruses with an average titer of $10^9$ TU/ml. On day 2 of differentiation, BM cells were harvested and subjected to spin-infection (90 min, 931$g$, 37°C) in the presence of polybrene at 8 µg/ml, at a multiplicity of infection of 2.5. 2 d later, the medium was changed and puromycin was added at a final concentration of 5 µg/ml. Cells were used on day 7 of differentiation. To generate the lentiviral expression construct pLVX Arf6-mCherry, the coding sequence of Arf6 was amplified using the primers Fw: 5′-GCCACCATGGGGAAGGTGCTA-3′ and Rv: 5′-TCACTTG-TACAGCTCGTCCAT-3′. mCherry was amplified with the primers Fw: 5′-GTGAGCAAGGGCGAGGAGGAT-3′ and Rv: 5′-TCACTTGTACAGCTCGTCCAT-3′. The lentiviral vector pLVX-EF1α-IRES-puro was linearized using SpeI and BamHI. The three fragments were assembled by the use of the In-Fusion HD Cloning Plus Kit (Takara Bio Europe) following the manufacturer's instructions. Plasmid DNA was then used to produce a lentivirus with a titer 5 × $10^9$ TU/ml. DC 2.4 cells were transduced in the same way as BM-DCs at a MOI of five and expanded in the presence of 5 µg/ml puromycin for five passages before puromycin removal. Transduction efficiency was verified by flow cytometry or by microscopy.

## Quantitative real-time PCR (qPCR)

BM-DCs on day 7 of differentiation were harvested, the pellets were collected, and the RNA was extracted using the NucleoSpin RNA XS kit (Macherey-Nagel). cDNA was obtained using the ImProm-II Reverse Transcription System kit (Promega), with random hexamers and 1 µg of total RNA. qPCR was performed with the SYBR Green method using Takyon ROX qPCR SYBR MasterMix blue dTTP (Eurogentec) and the following primers: GAPDH Fw: 5′-CCGTAGA-CAAAATGGTGAAGG-3′, GAPDH Rv: 5′-CGTGAGTGGAGTCATACTGGA-3′, Arf6 Fw: 5′-AGATCTTCGGGAACAAGGAAAT-3′, and Arf6 Rv: 5′-CACACGTTGAACTTGACGTTTT-3′. Relative gene expression of Arf6 was determined according to the $2^{(-\Delta\Delta CT)}$ method using the GAPDH housekeeping gene as a reference.

## Flow cytometry

BM-DCs on day 7 of differentiation were harvested and kept on ice water throughout the staining process. Where indicated, BM-DCs were acid stripped 120 s with 300 mM PBS-glycine pH 2.8, followed by neutralization by addition of cold medium, and two further washes with medium and PBS, respectively. Cells were then stained

in FACS buffer (2% FBS in PBS) with primary unconjugated antibodies, washed once in FACS buffer, stained with the appropriate mixture of secondary antibodies (donkey anti-mouse Alexa Fluor 488 or Streptavidin APC-Cy7), and Bv421-coupled CD11c antibodies for identification of DCs; cells were washed again and resuspended in FACS buffer before flow cytometry analysis. Acquisition was performed on a FACSCanto II (BD Biosciences). Dead cells were excluded by 7-AAD staining. Data analysis was performed using FlowJo (V10.1). The geometric mean fluorescence intensity of the CD11c⁺ cells plus the relevant marker was calculated. For intracellular staining, the FIX & PERM Kit (Life Technologies) was used according to the manufacturer's instructions. BFA decay experiments were performed as previously described (Montealegre et al, 2015). To measure anterograde transport, transduced C57BL/6 BM-DCs were acid stripped on ice, the acid was quenched, and the cells were reincubated in prewarmed medium at 37°C for the indicated duration in the presence or absence of 5 µg/ml BFA. Cells were then stained with AF6-biotin followed by streptavidin-Alexa488.

## Confocal microscopy

BM-DCs on day 6 of differentiation were seeded onto glass coverslips precoated with fibronectin overnight. The next day, cells were fixed with 2% paraformaldehyde, quenched with 300 mM glycine, and permeabilized using 0.2% saponin and 0.2% BSA in PBS. Primary antibodies were diluted in permeabilization buffer (1:2 for hybridoma supernatants, 1:100 EEA-1, 1:400 LAMP-1, 1:50 Rab11), and secondary antibodies to 1:200. After postfixation with 4% formaldehyde, quenching was carried out with 50 mM NH₄Cl before staining of nuclei with DAPI (100 ng/ml). For steady-state experiments, image acquisitions were performed with a 63× oil immersion objective (NA 1.4) through a laser scanning confocal microscope (TCS SP8-3X STED; Leica Microsystems). For kinetic experiments, Z-stacks of 2 µm were taken to visualize the entire volume of the cells in the field, with a spinning disk confocal microscope composed of a Yokogawa CSU-X1 spinning disk scanner, coupled to a Zeiss Observer Z1 inverted microscope, and controlled by Zen Blue software. Tile images were acquired with a Plan Apochromat 63× oil immersion objective (NA 1.46) through a Hamamatsu Orca Flash 4.0 sCMOS camera. For live cell imaging, cells were visualized through a single z-stack during the recording time, with an acquisition time of a picture every 30 s for up to 1 h, in a humidified chamber at 37°C and 5% CO2 supply. Images were processed with FIJI. Colocalization analysis was performed with JACoP from ImageJ, throughout the entire volume of each cell, unless otherwise specified. Data are reported as the Mander's coefficient.

## Antibody-mediated internalization assay

Transduced BM-DCs seeded onto coverslips were washed with 1× PBS, placed onto ice water, and then incubated with the respective antibody for 30 min. In case the cells were treated with acid, cells were extensively washed before adding the respective antibody. Excess antibodies were washed out, and then, the cells were incubated at 37°C for the indicated duration, without removing the

prebound cell surface antibodies. Cells were then fixed, permeabilized, and stained as described above.

### Quantification of 64-reactive MHC-I upon internalization

Transduced BM-DCs seeded onto coverslips were washed with 1× PBS and incubated 45' with 20 nM concanamycin B (ConB) (ab144228; Abcam) at 37°C. Cells were washed and acid-stripped as above. After extensive washing, mAb 64 was added to the cells for 30 min on ice, excess antibodies were washed, and the cells were fed with full medium containing 20 nM ConB during the chase. Finally, the cells were stained with antibodies against LAMP1, EEA-1, and a secondary antibody against internalized 64. Image acquisition was performed with the same laser power and gain for both shNT and shArf6 cells. After routine image processing, each cell was converted into a z-projection, the same threshold was applied for the color where 64 was evaluated, and finally converted into a mask. Using the embedded functions of FIJI, the area, the mean grey value, and the integrated density values were calculated per cell. The mean grey value represents the fluorescence intensity of 64, and the integrated density is the product of the area and the mean grey value, which corrects for potential variation in the size of the cells.

### Nocadazole treatment

DC2.4 cells transduced with Arf6-mCherry were treated with 5 μg/ml nocodozale (Ref 487928; Sigma-Aldrich) 30 min at 37°C or with DMSO solvent. Cells were fixed and stained for the indicated markers. As a positive control to confirm Golgi disruption, a staining for GM130 was included in nocodazole-treated cells imaged.

### Recycling assay

C57BL/6 BM-DCs on day 7 of differentiation were incubated with mAb AF6-biotin for 30 min on ice. After washing, the cells were incubated at 37°C for the indicated duration with 220 μM primaquine where indicated, as previously described (Ma et al, 2016). After internalization, cells were cooled to 4°C, washed, and acid-stripped. Then, aliquots were reincubated at 37°C, and the cells were finally labeled with streptavidin-AlexaFluor 488. The percentage of internalized H-2K$^b$ was calculated as (GeoMean after binding on ice – GeoMean after t = x internalization)/(GeoMean after binding on ice) × 100. The percentage of recycling K$^b$ was calculated as (GeoMean after t = x recycling – GeoMean after acid stripping)/(GeoMean after binding on ice – GeoMean after t = x internalization) × 100.

### Degradation assay

ICs between OVA-647 and an OVA antibody were prepared by mixing them at an equimolar ratio and incubating 1 h at 4°C. Transduced BM-DCs were then harvested and resuspended in a solution containing the ICs or soluble OVA-647, with the latter at the same concentration as in the IC, for 20 min on ice water. Cells were washed three times with cold PBS 1× and then incubated at 37°C in complete medium for the indicated time points. At each time point, cells were washed and placed on ice until the last time point. Finally, fluorescence was detected by flow cytometry. Although OVA-647 is not reported to be highly self-quenching, we have found that its degradation increased fluorescent brightness. Data are represented as a fold increase in fluorescence relative to the cells incubated at 4°C.

### Antigen presentation assays

Antigen cross-presentation assays were carried out essentially as described before (Merzougui et al, 2011). In brief, complexes formed between OVA and anti-OVA, P3UO and anti-CD11c, or P3UO and anti-CD206 were prepared by mixing the two respective components at an equimolar ratio and incubating for 1 h at 4°C. Then, the complexes were added to BM-DCs during 6 h, followed by fixation with 0.001% glutaraldehyde, washing with PBS-glycine 200 mM, and incubating with OT-I T cells at a ratio of 5 OT-I per DC. *Saccharomyces cerevisiae* cells expressing full length OVA at the cell surface (OVA-yeast) (Merzougui et al, 2011) were UV-irradiated and fed to the cells in a dilution starting from five yeast cells per DC. SL8 or GS-20 peptides were incubated for 4 h with the BM-DCs, which were then treated as above. To rule out extracellular antigen processing, cells prefixed using 0.004% glutaraldehyde were routinely included as controls. IL-2 secretion by the OT-I cells was used as a read-out of the experiments and measured by ELISA. Data are reported as the percentage relative to shNT-transduced cells at the maximal dilution of antigen.

### Statistics

In experiments where the variation in the absolute data between independent experiments masked fold changes between control and test samples, data were normalized relative to the control NT at 100%. Data reported as percentage relative to controls were evaluated with a one-sample test perfomed in GraphPad Software, under the null hypothesis that the column means of the sample are equal to 100%. The number of individual experiments is shown in each legend. Typical experiments are shown in supplementary figures. Microscopy data were evaluated with an unpaired non-parametric Mann–Whitney test, pooling individual cells from independent experiments per time point/marker. Data are significantly different if $P \leq 0.05$ (*), $P \leq 0.01$ (**), or $P \leq 0.001$ (***).

# Supplementary Information

# Acknowledgements

We are grateful to Paul Roche for preliminary recycling experiments. Lentiviruses were produced by the platform Structure Fédérative de Recherche Necker Vecteurs viraux et transfert de genes (Structure Féderative de Recherche Nêcker-Université Paris Descartes, S Fabrega). This work was supported by grant ANR-14-CE11-0014 of the Agence Nationale de Recherche.

## Author Contributions

S Montealegre: conceptualization, investigation, visualization, methodology, and writing—original draft.
A Abramova: investigation.
V Manceau: investigation.
A-F de Kanter: investigation.
P van Endert: conceptualization, data curation, formal analysis, supervision, funding acquisition, validation, project administration, and writing—review and editing.

## Conflict of Interest Statement

The authors declare that they have no conflict of interest.

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
