## [Reviewer comments · Life Science Alliance]

Life Science Alliance

The Role of MHC Class I Recycling and Arf6 in Cross-Presentation by Murine Dendritic Cells

Sebastian Montealegre, Anastasia Abramova, Valerie Manceau, Anne-Floor de Kanter, and Peter van Endert

DOI: <https://doi.org/10.26508/lsa.201900464>

Corresponding author(s): Peter van Endert, INSERM U1151

Review Timeline:

Submission Date:	2019-06-21
Editorial Decision:	2019-06-21
Revision Received:	2019-10-16
Editorial Decision:	2019-10-18
Revision Received:	2019-11-01
Accepted:	2019-11-04

Scientific Editor: Andrea Leibfried

Transaction Report:

Please note that the manuscript was previously reviewed at another journal and the reports were taken into account in the decision-making process at Life Science Alliance.

Referee #1

Report for Author:

In a manuscript entitled "Revisiting the role of MHC class I recycling and Arf6 in crosspresentation by murine dendritic cells", Montealegre and colleagues examine the role played by Arf6 in the trafficking of MHC class I molecules in DCs. The manuscript is generally well-written. The results are generally quite clear, supported by convincing immunofluorescence data (mostly). The data obtained indicate that Arf6 is not required for the transport of MHC I to and the recycling from the plasma membrane. They also provide evidence that Arf6 is not involved in the transport of these molecules to the Rab11+ endocytic recycling compartment, a result that contradict a previous report.

Although the results are in general convincing (the results showing a role of ARF6 in crosspresentation of ova internalized through the Fc receptor are interesting), the overall impression is that they are largely descriptive and provide only a marginal increment in the understanding of the molecular mechanisms and pathways responsible for the handling and recycling of MHC class I molecules in DC, as well as the role of Arf6 in crosspresentation. These data would be perhaps suited for a more specialized journal of immunology.

Referee #2 Review

Report for Author:

The authors present data showing that knockdown of Arf6 in BMDC results in increased cell surface expression of folded and unfolded MHC-I, and demonstrate that the increase in surface MHC-I is not a result of reduced internalization or recycling. They suggest that this increase might be attributed to reduced lysosomal degradation of MHC-I in Arf6 knockdown cells. Data is presented suggesting that Arf6 localizes to Rab11 positive organelles. However, internalized MHC-I does not localize to Rab11 positive compartment. Knockdown of Arf6 only impacts cross-presentation of antigens delivered as immune complexes. Overall, the data constitute an intriguing set of observations but it remains unclear what underlying mechanisms they reflect, or even whether they even reflect the same or related processes. This makes it questionable whether this journal is the right journal for the manuscript.

Major concerns:

1. In Fig. 3 the authors perform antibody-mediated MHC-I internalization to determine the kinetics of internalization and recycling, raising the concern that the antibodies might drive altered kinetics and trafficking of MHC-I. It would be better to use an assay that does not involve bound antibody, such as surface biotinylation.
2. Arf6 localizes to a Rab11-positive compartment. The perinuclear region of the cell contains lysosomes, TGN and ERC, and it is difficult to determine the colocalization of markers with great precision. Treatment of cells with Nocodazole for 30min disperses all the perinuclear compartments to the periphery of the cell as distinct units, and allows for a more accurate assessment of the distribution of markers. The colocalization data presented in Figs. 2 and 6 may be improved by a short Nocodazole treatment.
3. Authors demonstrate that Arf6 knockdown cells poorly cross-present the OVA antigen delivered as immune complexes and that this does not correlate with the increased surface MHC-I observed. It would be useful to compare the rates of degradation of IC-OVA in cells transduced with Arf6 shRNA or a non-targeting shRNA. It would also be useful to perform live cell imaging of cells internalizing DQ OVA by a route other than IC (EV figure 6, video1 and 2).

Minor points

- 1) Figure 4G, second panel, there is a typo in the Y-axis label.

June 21, 2019

Re: Life Science Alliance manuscript #LSA-2019-00464-T

Prof. Peter van Endert
INSERM U1151
149 rue de Sèvres
Paris 75015
FRANCE

Dear Dr. van Endert,

Thank you for transferring your manuscript entitled "Revisiting the Role of MHC Class I Recycling and Arf6 in Cross-Presentation by Murine Dendritic Cells" to Life Science Alliance. The manuscript was assessed by expert reviewers at another journal before, and the editors transferred those reports to us with your permission.

The reviewers appreciated your results and found them robust. They would, however, have expected further reaching insight, lack of which does not preclude publication in Life Science Alliance. We would thus like to invite you to submit a revised version to us based on the reviewer reports already at hand. While a surface biotinylation assay is not needed, the other specific comments of reviewer #2 should get addressed in a revised version. Please also provide a full point-by-point response to both reports.

Thank you for this interesting contribution to Life Science Alliance. We are looking forward to receiving your revised manuscript.

Sincerely,

Andrea Leibfried, PhD
Executive Editor
Life Science Alliance
Meyerhofstr. 1
69117 Heidelberg, Germany
t +49 6221 8891 502
e a.leibfried@life-science-alliance.org

B. MANUSCRIPT ORGANIZATION AND FORMATTING:

Referee #1:

In a manuscript entitled "Revisiting the role of MHC class I recycling and Arf6 in cross-presentation by murine dendritic cells", Montealegre and colleagues examine the role played by Arf6 in the trafficking of MHC class I molecules in DCs. The manuscript is generally well-written. The results are generally quite clear, supported by convincing immunofluorescence data (mostly). The data obtained indicate that Arf6 is not required for the transport of MHC I to and the recycling from the plasma membrane. They also provide evidence that Arf6 is not involved in the transport of these molecules to the Rab11+ endocytic recycling compartment, a result that contradict a previous report.

Although the results are in general convincing (the results showing a role of ARF6 in cross-presentation of ova internalized through the Fcγ receptor are interesting), the overall impression is that they are largely descriptive and provide only a marginal increment in the understanding of the molecular mechanisms and pathways responsible for the handling and recycling of MHC class I molecules in DC, as well as the role of Arf6 in crosspresentation. These data would be perhaps suited for a more specialized journal of immunology.

Although we did not contest the editorial decision at the other journal, we disagree with this reviewer. We believe that rather than representing a marginal increment, our data correspond to a complete revisiting of the understanding of class I recycling in dendritic cells. The literature contains numerous manuscripts and reviews stating an involvement of MHC-I recycling in cross-presentation, with a role for an Arf6-dependent pathway, assumptions contradicted by our findings.

Referee #2:

The authors present data showing that knockdown of Arf6 in BMDC results in increased cell surface expression of folded and unfolded MHC-I and demonstrate that the increase in surface MHC-I is not a result of reduced internalization or recycling. They suggest that this increase might be attributed to reduced lysosomal degradation of MHC-I in Arf6 knockdown cells. Data is presented suggesting that Arf6 localizes to Rab11 positive organelles. However, internalized MHC-I does not localize to Rab11 positive compartment. Knockdown of Arf6 only impacts cross-presentation of antigens delivered as immune complexes. Overall, the data constitute an intriguing set of observations but it remains unclear what underlying mechanisms they reflect, or even whether they even reflect the same or related processes. This makes it questionable whether this journal is the right journal for the manuscript.

Major concerns:

1. In Fig. 3 the authors perform antibody-mediated MHC-I internalization to determine the kinetics of internalization and recycling, raising the concern that the antibodies might drive altered kinetics and trafficking of MHC-I. It would be better to use an assay that does not involve bound antibody, such as surface biotinylation. As agreed with the editor and although it is correct that using a biotinylation assay could corroborate our findings, we did not perform experiments using biotinylation in the revised paper, as this would have required substantial setup efforts.

2. *Arf6 localizes to a Rab11-positive compartment. The perinuclear region of the cell contains lysosomes, TGN and ERC, and it is difficult to determine the colocalization of markers with great precision. Treatment of cells with Nocodazole for 30min disperses all the perinuclear compartments to the periphery of the cell as distinct units, and allows for a more accurate assessment of the distribution of markers. The colocalization data presented in Figs. 2 and 6 may be improved by a short Nocodazole treatment.*

Thank you for this suggestion. We have performed confocal microscopy studies on cells treated for 30 min with nocodazole, as proposed by the reviewer. The results are shown in the new **Fig. EV2** and comprise images for the corresponding to the co-stainings in Fig. 2 and Fig. 6. As expected, the nocodazole shock inhibits assembly of the Golgi apparatus and disperses or retains in the cellular periphery structures staining for EEA-1, Lamp1, Arf6 and MHC class I. Interestingly, upon disassembly of microtubules, peripheral Arf6 structures maintain some co-localization with early (EEA-1+) and late (Lamp1+) endosomes but fail to colocalize with Rab11 or MHC class I.

3. *Authors demonstrate that Arf6 knockdown cells poorly cross-present the OVA antigen delivered as immune complexes and that this does not correlate with the increased surface MHC-I observed. It would be useful to compare the rates of degradation of IC-OVA in cells transduced with Arf6 shRNA or a non-targeting shRNA.*

We have performed a FACS-based degradation assay using soluble OVA-647 and complexes of OVA-647 with an OVA antibody. The results are shown in **figure 9F and 9G**, respectively, and the corresponding histograms in **figure EV6**. The assay is based on the principle that upon exposure to acidic pH, equivalent to degradation, the fluorescence will increase. As shown in the previous version of the manuscript, when fluorescent OVA is incubated at 4°C with specific antibodies (**figure EV6C**), OVA fluorescence increases, likely due to steric effects since the pH remains neutral at this step. However, as shown in **figure EV6AB**, upon incubation at 37°C the fluorescence increases, indicating OVA exposure to acid pH. We find that Arf6 knockdown does not affect that the rate of degradation of soluble OVA-647. This contrasts with the degradation of immune complexes that is somewhat accelerated in the absence of Arf6.

It would also be useful to perform live cell imaging of cells internalizing DQ OVA by a route other than IC (EV figure 6, video 1 and 2).

We have performed live cell imaging using soluble DQ-OVA, recording two videos from independent experiments of 1h length (**Videos 6 and 7**). Flow cytometric evaluation shows that soluble DQ-OVA is degraded intracellularly (**figure EV6 A**), starting at 30 minutes of internalization. Interestingly, the relative intracellular movement of OVA-containing vesicles and the central Arf6 compartment differs strikingly between cells fed soluble DQ-OVA and cells fed immune complexes.

Minor points

1) *Figure 4G, second panel, there is a typo in the Y-axis label.*

We have corrected the typo.

Editor

Yes, addressing points 2 and 3 of reviewer #2 will be acceptable. The potentially altered kinetics of endocytosis and trafficking in the antibody-based endocytosis assay should then get discussed.

We discuss this issue on pages 18/19 of the revised manuscript.

...and it would perhaps be good to show higher magnifications of the cells to more easily appreciate the localization/co-localization.

We now show higher magnifications of the cells in all panels of **Figures 2, 3, 4, 5** and **EV2**.

October 18, 2019

RE: Life Science Alliance Manuscript #LSA-2019-00464-TR

Prof. Peter van Endert
INSERM U1151
149 rue de Sèvres
Paris 75015
France

Dear Dr. van Endert,

Thank you for submitting your revised manuscript entitled "The Role of MHC Class I Recycling and Arf6 in Cross-Presentation by Murine Dendritic Cells". I have re-assessed the revised version and your point-by-point response and I appreciate the introduced changes. I would thus be happy to publish your paper in Life Science Alliance pending final revisions necessary to meet our formatting guidelines:

- please note that we only have supplementary figures at Life Science Alliance, please re-name your EV figures into S figures
- please add a callout in the manuscript text to Fig 3E and S4 (currently EV4)
- please list 10 authors et al in your reference list
- please add information on stat tests and p values within the figures legends, too
- please include information in your figure legends for abbreviations (eg. "NT" for control condition in KD experiments)
- please include information on the nuclear staining seen in the merge panels in all figure legends

A. FINAL FILES:

-- High-resolution figure, supplementary figure and video files uploaded as individual files: See our detailed guidelines for preparing your production-ready images, <http://www.life-science->

alliance.org/authors

B. MANUSCRIPT ORGANIZATION AND FORMATTING:

Sincerely,

Andrea Leibfried, PhD
Executive Editor
Life Science Alliance
Meyershofstr. 1
69117 Heidelberg, Germany
t +49 6221 8891 502
e a.leibfried@life-science-alliance.org

November 4, 2019

RE: Life Science Alliance Manuscript #LSA-2019-00464-TRR

Prof. Peter van Endert
INSERM U1151
149 rue de Sèvres
Paris 75015
France

Dear Dr. van Endert,

Thank you for submitting your Research Article entitled "The Role of MHC Class I Recycling and Arf6 in Cross-Presentation by Murine Dendritic Cells". It is a pleasure to let you know that your manuscript is now accepted for publication in Life Science Alliance. Congratulations on this interesting work.

*****IMPORTANT:** If you will be unreachable at any time, please provide us with the email address of an alternate author. Failure to respond to routine queries may lead to unavoidable delays in publication.*******

DISTRIBUTION OF MATERIALS:

Again, congratulations on a very nice paper. I hope you found the review process to be constructive and are pleased with how the manuscript was handled editorially. We look forward to future exciting submissions from your lab.

Sincerely,
